# Gene–Toxicant Interactions in Gulf War Illness: Differential Effects of the *PON1* Genotype

**DOI:** 10.3390/brainsci11121558

**Published:** 2021-11-25

**Authors:** Jacqueline Vahey, Elizabeth J. Gifford, Kellie J. Sims, Blair Chesnut, Stephen H. Boyle, Crystal Stafford, Julie Upchurch, Annjanette Stone, Saiju Pyarajan, Jimmy T. Efird, Christina D. Williams, Elizabeth R. Hauser

**Affiliations:** 1Cooperative Studies Program Epidemiology Center-Durham, Durham VA Medical Center, Durham VA Health Care System, Durham, NC 27705, USA; jacqueline.vahey@duke.edu (J.V.); Elizabeth.Gifford14@va.gov (E.J.G.); kellie.sims@va.gov (K.J.S.); Blair.Chesnut@va.gov (B.C.); Stephen.Boyle@va.gov (S.H.B.); Crystal.Stafford@va.gov (C.S.); Julie.Upchurch@va.gov (J.U.); Jimmy.Efird@va.gov (J.T.E.); Christina.Williams4@va.gov (C.D.W.); 2Computational Biology and Bioinformatics Program, Duke University School of Medicine, Durham, NC 27705, USA; 3Center for Child and Family Policy, Duke Margolis Center for Health Policy, Duke University Sanford School of Public Policy, Durham, NC 27708, USA; 4Pharmacogenomics Analysis Laboratory, Research Service, Central Arkansas Veterans Healthcare System, Little Rock, AR 72205, USA; Annjanette.Stone@va.gov; 5Massachusetts Veterans Epidemiology Research and Information Center (MAVERIC), VA Boston Healthcare System, Boston, MA 02130, USA; Saiju.Pyarajan@va.gov; 6Duke Molecular Physiology Institute, Department of Biostatistics and Bioinformatics, Duke University Medical Center Durham, NC 27701, USA

**Keywords:** veteran health, Gulf War illness, GWI, pesticides, pyridostigmine bromide, gene–environment interactions, *PON1*

## Abstract

About 25–35% of United States veterans who fought in the 1990–1991 Gulf War report several moderate or severe chronic systemic symptoms, defined as Gulf War illness (GWI). Thirty years later, there is little consensus on the causes or biological underpinnings of GWI. The Gulf War Era Cohort and Biorepository (GWECB) was designed to investigate genetic and environmental associations with GWI and consists of 1343 veterans. We investigate candidate gene–toxicant interactions that may be associated with GWI based on prior associations found in human and animal model studies, focusing on SNPs in or near *ACHE*, *BCHE*, and *PON1* genes to replicate results from prior studies. *SOD1* was also considered as a candidate gene. CDC Severe GWI, the primary outcome, was observed in 26% of the 810 deployed veterans included in this study. The interaction between the candidate SNP rs662 and pyridostigmine bromide (PB) pills was found to be associated with CDC Severe GWI. Interactions between PB pill exposure and rs3917545, rs3917550, and rs2299255, all in high linkage disequilibrium in *PON1*, were also associated with respiratory symptoms. These SNPs could point toward biological pathways through which GWI may develop, which could lead to biomarkers to detect GWI or to better treatment options for veterans with GWI.

## 1. Introduction

Gulf War veterans began reporting chronic systemic symptoms almost immediately after returning from deployment to the Gulf in support of the 1990–1991 Gulf War [1,2,3,4]. These symptoms have only increased, in both severity and frequency, over the past 30 years, and have been defined as Gulf War illness (GWI) [1,5,6,7,8,9,10]. The 2016 Institute of Medicine (IOM) report on GWI indicated great heterogeneity in the expression of GWI across the veteran population, suggesting heterogeneity in genetic susceptibility, environmental exposures, and/or their interactions [11]. There are still important gaps in our understanding of genetic or environmental risk factors for GWI, especially in how these risk factors may interact to increase susceptibility to GWI. Biological understanding of GWI is necessary to discover and design better treatment options for veterans with GWI.

Summarizing several decades of research on GW exposures, the VA Gulf War Research Advisory Council identified pyridostigmine bromide (PB) pills and the use of pesticides as being consistently associated with chronic multisymptom illness in multivariable adjusted models [12]. These consistently observed associations with PB and pesticide exposures suggest biological pathways, enzymes and genes for consideration in genetic analysis. Possible genetic risk loci have been identified over the past 25 years of GWI research, although replication of these results has been difficult. Butyrylcholinesterase was identified by Steele et al., 2015, where functional variants with decreased efficiency were associated with higher risk of Kansas-definition GWI [13]. Haley et al., 1999, identified an association between low arylesterase activity in paraoxonase-1 and higher risk of Haley Symptom Complex 2, Confusion-ataxia [14]. Association between case status and *PON1*-amino acid 192 R variant was also reported [14]. Inhibition of acetylcholinesterase function is often cited as an explanation for why the pesticides, PB, and nerve agents are suspected to be in the causal pathway of GWI [7,12,15,16,17,18,19,20,21,22]. Rodent models for GWI have been developed using combinations of pesticides, PB, sarin surrogate diisopropyl fluorophosphate (DFP), and chronic stress or corticosterone [15,16,17,19,23,24,25,26,27,28,29].

The enzymes acetylcholinesterase, butyrylcholinesterase, and paraoxonase are encoded by the genes *ACHE*, *BCHE*, and *PON1*, respectively, which have been studied as candidate genes underlying variation in response to GW exposures. In addition, *SOD1* is a gene associated with amyotrophic lateral sclerosis (ALS), which has been associated with GW service [30,31,32,33] and is also postulated to be involved in gene-by-environment interactions [34,35]. The genetic analysis of these candidate genes in GWI has been limited to association studies of a small number of functional polymorphisms in each gene. Common functional polymorphisms in both *BCHE* and *PON1* genes have been associated with GWI in prior studies [13,14]. The atypical butyrylcholinesterase variant, caused by the genetic variant rs1799807, is known to cause slower metabolism of succinylcholine, a common neuromuscular blockade used in surgery [36]. In a small sample of US GW veterans, Steele et al., 2015 show association of butyrylcholinesterase variants with GWI with exposure to PB pills [13]. Steele et al., 2015 considered the wild type (U), atypical (A; rs1799807), Kalow (K), atypical-Kalow (AK), and fluoride (F) variants, comparing veterans with K/K, U/AK, U/A, A/F, and AK/F genotypes to veterans with U/K genotypes and to veterans with U/U genotypes. The functional variant in *PON1* discussed by Haley et al., rs662, has two substrate-dependent, differentially active phenotypes: one, the 192Q variant, has higher catalytic efficiency for sarin, soman, and diazoxon (the A allele at this locus leads to the Q version of the protein), while the other, the 192R variant, has higher catalytic efficiency for paraoxon (G allele at this locus leads to the R version of the protein) [37]. The common SNP rs662 has also been associated with increased risk for ALS and Pon1 activity has been associated with cardiovascular disease [38,39,40,41]. The relationships between genotype, catalytic activity, and disease outcomes are complex and likely involve additional variants in the *PON1* gene [42]. Rare functional variants in human acetylcholinesterase (*ACHE*) have been implicated in anti-acetylcholinesterase sensitivity [43]. However, common variants have not been associated with acetylcholinesterase activity. One common variant, rs1799805, is responsible for the YT blood type, which may be in linkage disequilibrium with rare variants. Our goal in this gene-by-environment interaction study is to evaluate these specific candidate variants as well as assess common variation across the genes *SOD1*, *ACHE*, *BCHE*, and *PON1* for association with GWI. Our hypothesis is that GWI case status and features of GWI and, specifically, the CDC Severe GWI case definition will be associated with the interaction between these genes and PB pill or pesticide exposures among deployed GW veterans.

## 2. Materials and Methods

### 2.1. Introduction to GWECB

The Gulf War Era Cohort and Biorepository (GWECB) consists of DNA, electronic medical records, and survey data from individuals who served in the United States Armed Forces during the 1990–1991 Gulf War [44]. Data collected includes self-reported exposures, exposure lengths, symptoms, symptom severity ratings, clinical conditions, and deployment locations. Women, veterans of color, and deployed veterans were oversampled, and veterans did not have to use the Veterans Affairs system for healthcare to be eligible for inclusion.

### 2.2. GWI Phenotypes

Veterans from the GWECB cohort have been classified as meeting or not meeting the case definition using the CDC, CDC Severe, Kansas full case definition, and Kansas symptom criteria [6]. Case status was coded using a deterministic SAS algorithm and basic survey data cleaning was also performed in SAS 9.4, as described in Vahey et al. and Gifford et al. [6,45].

Eleven outcomes were used for the regression analysis: the CDC Severe GWI Case Definition, the three CDC Severe symptom domains (CDC Severe: Musculoskeletal, CDC Severe: Mood/Cognition, and CDC Severe: Fatigue), Kansas Symptom Criteria, and the six Kansas symptom domains (Kansas: Pain, Kansas: Fatigue, Kansas: Respiratory, Kansas: Skin, Kansas: Mood/Neurological/Cognitive, and Kansas: Gastrointestinal). An additional variable derived from the GWI case algorithm is the Kansas exclusionary conditions indicator. This variable identifies veterans with at least one of 21 exclusionary health conditions, such as diabetes, heart disease, and cancer. Fulfilling the exclusionary criterion precludes an individual from being considered a GWI case. The CDC Severe and Kansas GWI case definitions are known to be heterogeneous; it is possible that multiple traits with different underlying etiologies make up GWI, complicating analysis. Thus, studying the symptom domains separately may isolate more homogeneous subphenotypes. Each symptom domain in the CDC Severe and Kansas case definitions were considered as individual outcomes, as in prior studies [13,14]. Previous analyses of deployment in GWECB demonstrated a strong and consistent association of deployment with the CDC Severe Case Definition and thus CDC Severe will serve as the primary outcome for this analysis [6].

### 2.3. Gulf War Exposures

Veterans deployed to the Gulf during the 1990–1991 Gulf War were asked to report on their exposures. The survey asks for both a yes/no response and a “number of days exposed” response for 11 exposure questions. Appendix A Figure A1 is a scan of the survey page for these exposure responses. Ambiguous values were coded based on previous work [6]: number of days exposed was set to zero for individuals who said they did not experience an exposure, regardless of their secondary answer. Exposures used were “Took pyridostigmine bromide (anti-nerve agent pills)”, “Used pesticide cream or liquid on your skin”, “Wore a uniform treated with pesticides”, and “Used insect baits/no-pest strips in your living area”. The PB exposure variable used was the number of days each individual reported being exposed to PB pills; the options were “No, none”, “Yes, 1–6 days”, “Yes, 7–10 days”, and “Yes, 31 days or more”. The pesticide exposure variable combined the number of days reported for all three pesticide variables listed above, based on the minimum number of days per option (i.e., “Yes, 1–6 days” was counted as 1 day). Less than 31 days combined exposure was a short exposure, 31–60 days combined exposure was a medium exposure, and more than 60 days combined exposure was a long exposure. Pesticide use and PB pill exposure were chosen as candidate exposures over “exposed to biological or chemical warfare agents”, following conclusions of the 2014 Research Advisory Committee report [12].

### 2.4. Genotypes and Genetic Data Cleaning

Blood samples were collected for DNA extraction, as described by Khalil et al. [44]. DNA samples were whole-genome amplified, hybridized to microarrays, single-base extended and stained using the Automated Protocol for the Illumina Infinium HTS Assay in batches of two 96-sample plates per run. Genotyping was done on an Illumina Omni2.5-8v1.4 microarray at the Pharmacogenomics Analysis Laboratory (PAL) in the Central Arkansas Veterans Healthcare System. Analysis of agreement in genotypes between sample duplicates indicated excellent genotyping quality. Plink 1.9 was used for genetic data cleaning and analysis [46]. A total of 2,372,461 SNPs were genotyped; 82,369 SNPs were removed due to missingness greater than 1% and 512,116 SNPs with minor allele frequency (MAF) less than 1% were removed, including the two SNPs included by Steele et al. [13], leaving 1,777,976 SNPs. From those 1,777,976 SNPs, 3 SNPs were chosen for Tier 1 candidate SNP testing (i.e., *BCHE, PON1, ACHE*) and 371 SNPs were chosen for Tier 2 candidate SNP testing, representing all SNPs +/− 50 kb from the annotated gene. Individuals who returned surveys but who did not provide a specimen for genetic testing were excluded from analysis. Beginning with 1343 samples, replicates were removed, keeping the most informative sample for each individual. Two individuals were removed because of missing genotypes and 13 individuals were removed because of perceived genetic relatedness, leaving 1260 veterans for genetic analysis. As only veterans deployed to the Gulf were asked to report on their exposures, those who were deployed elsewhere and those who were not deployed were removed from the analysis. A total of 810 GW-deployed veterans were in the final analytic dataset after genotype and phenotype data cleaning, as seen in Figure 1.

### 2.5. Statistical Analysis

We fit a gene/environment interaction model to estimate the effect and statistical significance of specific candidate gene/toxicant interactions. We used logistic regression as implemented in PLINK 1.9 to test the effect of genetic variants and gene-by-exposure interaction [47]. As discussed above, the primary outcome was the CDC Severe phenotype. Analyses of other GWI case phenotypes and the individual symptom domain phenotypes were considered to be secondary. Given the complexity of the GWI phenotypes and the correlation among individual symptom domains, as well as their relationship to GWI outcomes, we did not adjust for multiple comparisons across the phenotypes but rather reported the *p*-values and effect sizes for each analysis.

Ten genetic principal components, representing population structure, along with age and sex were used as covariates for the basic model. The fully adjusted model included age, sex, education, income, and a binary indicator of the Kansas exclusionary criterion as an indicator of presence or absence of medical conditions. Because the Kansas exclusionary criterion is part of the full Kansas GWI case definition, the Kansas symptom criteria indicator was used as an outcome rather than the full Kansas GWI case definition in the fully adjusted model. Equation (1) outlines the basic model and Equation (2) outlines the fully adjusted model. We tested *β_3_* using the standard allele test in an additive genetic model as implemented in PLINK.
(1)$$Y=\beta_{0}+\beta_{1}*SNP+\beta_{2}*\;E\;+\beta_{3}*SNP*E+\beta_{4}*\;\mathrm{age}\;+\beta_{5}*\;\mathrm{sex}$$
(2)$$Y=\beta_{0}+\beta_{1}*SNP+\beta_{2}*\;E\;+\beta_{3}*SNP*E+\beta_{4}*\;\mathrm{age}\;+\beta_{5}*\;\mathrm{sex}\;+\beta_{6}*\;\mathrm{education}+\;\beta_{6}\;*\;\mathrm{income}+\beta_{7}*\;\mathrm{Kansas}\_\mathrm{Exclusion}$$

Interaction trends were tested for significance using the Cochran–Armitage test for trend, according to the *prop_trend_test()* function in the *rstatix* package in R version 3.61. LocusZoom plots were generated using legacy LocusZoom.org [48]. Tier 1 SNPs were tested at *p* = 0.05, while Tier 2 SNPs were tested at *p* = 0.00013 using a Bonferroni correction for the 371 SNPs in Tier 2. SNPs were prioritized based on *p*-value and representation across phenotypes.

### 2.6. SNP Selection

Both models were performed for each of the 11 outcomes, and both exposures. For inclusion in these models, we identified two kinds of *a priori* SNPs based on published evidence for gene–toxicant interactions in GWI. Tier 1 SNPs in Table 1 have been previously identified in genetic analysis of GWI. Tier 2 SNPs, enumerated in Table 1, were chosen to cover all common variation in the genes of interest.

## 3. Results

### 3.1. Demographics and Outcomes

Veterans who fulfill the CDC Severe GWI criteria and those who do not are similarly distributed by sex, age group, and OEF/OIF deployment. Those who were Black or Hispanic, in the Army, have a lower household income, or were deployed active duty were more highly represented among those who fulfill the CDC Severe GWI criteria than those who do not fulfill the CDC Severe GWI criteria (Table 2). Those who earned advanced degrees or who were in the reserves were underrepresented among those who fulfill the CDC Severe GWI criteria.

Across the 810 deployed veterans, 26% were categorized as fulfilling the CDC Severe GWI criteria. CDC Severe case status was associated with PB and pesticide exposure (Table 3). Table 3 shows that 36% of those who report long PB exposure (*n* = 149) and 33% of those who report long pesticide exposure (*n* = 156) reported symptoms consistent with CDC Severe GWI compared to 18% and 19% of those who reported no PB or no pesticide exposure, respectively. Across all outcomes, a larger proportion of those who were exposed to either pesticides or PB pills fulfilled the symptom criteria compared to those in the overall deployed group (Appendix A Table A1 and Table A2).

### 3.2. Association of Tier 1 SNP/Exposure Interactions with Primary and Secondary Outcomes

The results for association for interactions of PB and pesticide exposure with the Tier 1 candidate SNPs with the primary outcome (CDC Severe GWI) and the secondary outcomes are shown for the covariate adjusted model (Table 4). Only one SNP, rs662, showed a statistically significant interaction with PB exposure for the CDC Severe GWI outcome (*p* = 0.049). Among the secondary outcomes, the interaction between rs662 (*PON1*) and pesticides had a *p*-value of 0.029 in the fully adjusted model of the Kansas Gastrointestinal symptom domain. The other Tier 1 interaction below *p* = 0.05 was rs1799805 (*ACHE*) with PB pill exposure for the Kansas skin symptom domain. Rs1799807 (*BCHE*) and rs1799805 (*ACHE*) had low minor allele frequencies: 0.022 and 0.039, respectively, likely resulting in lower power to detect differences across groups for these two variants. LocusZoom plots for these results can be found in Appendix A Figure A2, Figure A3, Figure A4 and Figure A5.

Figure 2 illustrates the unadjusted CDC Severe GWI frequencies across rs662 genotype (AA—homozygote 192Q phenotype, AG—heterozygote and GG—homozygote 192R phenotype) by PB pill length of exposure group. Significant trends were observed for longer lengths of exposure with higher rates of GWI were seen in the *PON1* rs662 GA and GG genotypes, with *p*-values for the simple test for trend of 0.0004 and 0.0084, respectively. This trend was not observed in the AA group (*p* = 0.179).

### 3.3. Association of Tier 2 SNP/Exposure interactions with CDC Severe GWI

Common SNPs in the four candidate genes were plotted by genomic location using LocusZoom, highlighting the SNP with the lowest exposure/SNP interaction *p*-value for each gene. The LocusZoom plots for association of the interaction between PB and variants in *ACHE*, *BCHE, PON1*, and *SOD1* for the primary outcome, CDC Severe GWI, are shown in Figure 3. Similar plots for the interaction between pesticide exposure and variants in *ACHE*, *BCHE*, *PON1*, and *SOD1* are included in Appendix A Figure A6. No results achieved Bonferroni-corrected statistical significance. The lowest *p*-value among the PB exposure/SNP interaction terms is in *PON1* for the SNP rs2299260 (*p* = 0.005) (Table 5). Likewise, the lowest *p*-value for the interaction term with pesticides is in *PON1* for rs62467349 (*p* = 0.0086) for CDC Severe GWI (Table 5).

### 3.4. Association of Tier 2 SNP/Exposure Interactions with Secondary Outcomes

Across the secondary outcomes, three SNPs showed statistically significant interactions with PB exposure for the Kansas Respiratory Symptom Domain (rs3917545, rs3917550, and rs2299255). These three SNPs were in high linkage disequilibrium with one another (*r^2^* = 1), across all ancestry groups and are in the *PON1* gene. Other gene/environment interaction associations with individual symptom domains were nominally significant (*p* < 0.05), although given the small sample size and lack of power, many are likely to be false positive signals or a result of correlation among the symptoms themselves. The linkage disequilibrium between the SNPs identified in *PON1* is variable with moderate LD between rs662 and rs3917545, rs3917550, and rs2299255 in European ancestry and low LD between the same SNPs in African ancestry (*r*^2^ = 0.036 and *r*^2^ = 0.39 for YRI and CEU, respectively, Appendix A Figure A7 and Figure A8 [52]).

The group of interactions that was significant in the Kansas Respiratory domain model remained significant according to both the basic and fully adjusted models, but only in the PB interaction model. Figure 4 shows the interaction *p*-values for the region plotted on the −log_10_ scale. The *PON1* candidate SNP rs662 is not in high LD (*r*^2^ < 0.4) with rs3917545.

## 4. Discussion

We identified a statistically significant association between CDC Severe GWI and the rs662/PB pill interaction, as seen in Figure 2, representing a pattern consistent with a gene–environment interaction model. While this was not the lowest *p*-value interaction in the study, it is the only signal that appears for our primary outcome (i.e., CDC Severe GWI) out of the Tier 1 SNPs. Significantly higher rates of CDC Severe GWI with increasing exposure to PB pills were detected for GG and GA genotypes. This increase in risk does not appear in the AA genotype, indicating that exposure alone is not responsible for this effect and separating the risk genotypes from the non-risk genotype. A significant trend among those who reported over 31 days exposure to PB pills suggests an additive risk according to the count of G alleles. This is consistent with previous reports in the literature, that the R version of the paraoxonase enzyme is less effective in detoxifying most substrates, and therefore results in higher risk with more exposure [51,53]. Rs662 is a functional variant; the rs662 G alternative allele causes a change from a glutamine to an arginine at amino acid 192. As early as 1999, Haley et al. demonstrated a genotype-specific difference in paraoxonase (arylesterase) activity which was strongly associated with GWI [14]. The Q version of the protein was previously found to have higher catalytic activity across most substrates, including sarin, while the R version of the enzyme was found to more efficiently detoxify paraoxon. In genome-wide association analyses, two variants in *PON1* were associated at genome-wide significance level with paraoxonase enzyme activity, rs854572 and rs2057681, with rs2057681 in complete linkage disequilibrium (*r*^2^ = 1) with rs662 in European Americans [52,54]. Where prior studies have measured catalytic activity, we measured genotype using a genome-wide array. The multiple associated variants that we observed in *PON1* were not in high LD with one another, reflecting the conclusions of Jarvik et al. that multiple independent variants are associated with arylesterase activity of *PON1* [42]. Follow-up analysis to assess enzymatic activity has not been completed in GWECB and would be required to fully replicate this finding. This variant has also been implicated in several other diseases, including as a modifier of sporadic ALS [39], which affects Gulf War veterans at higher rates than other comparable military groups [32].

Analysis of common variants within the candidate genes also identified *PON1* variants as interacting with GW exposures. Rs3917545 was statistically significant after Bonferroni correction interacting with PB exposure for association with the Kansas Respiratory symptom domain outcome. Two additional SNPs were also significant, are all in linkage disequilibrium with one another, and thus are likely a single signal. These SNPs are all non-coding variants that have not been identified by other GWI studies, although one of the SNPs, rs3917545 has been identified as associated with blood protein levels of erythroid membrane associated protein in a plasma proteome screen [55,56]. Additional *PON1* SNP-exposure pairs had suggestive *p*-values (*p* < 0.05) but did not meet the threshold for statistical significance. Whether these represent independent signals or are merely correlated with other SNPs, exposures and/or outcomes will require a larger dataset to untangle. The other three candidate genes *BCHE*, *ACHE*, and *SOD1* did not yield any statistically significant interactions for this candidate gene–environment interaction study. Larger datasets will be required for additional discovery of gene–environment interactions related to GWI.

There are several limitations to our study. While our sample size is large relative to other studies of GWI with genetic information, we had limited power to detect rare variants, particularly those in the *BCHE* gene. We were also unable to replicate the *BCHE*/PB interaction identified by Steele et al. [13]. This is partially due to the lower allele frequency of the *BCHE* variants; Steele et al. combined several BCHE variants together for analysis, but we did not test variants that appeared in our data at less than 1% minor allele frequency. We prioritized candidate SNPs into two tiers, which allowed us to identify several significant signals, one of which has biological meaning. Given our sample size, we cannot dismiss type I error as a possible explanation for our findings. Future work could examine data from the MVP study, including over 100,000 Gulf War Era veterans with genotypes [57], which would allow for greater statistical power to detect gene-by-environment interactions. Future work could also analyze catalytic activity of our candidate genes, specifically Pon1, which could increase confidence in these findings. An additional limitation is around the generalizability of our findings. The GWECB survey had a very low return rate and may not be representative of the general Gulf War veteran population [44]. An important assumption in our model is the independence of genotype and exposure. While we did not observe an association between genotype and exposures, we are unable to fully rule out a possible dependency. Genotype proportions by exposure time remained consistent across all exposure categories, indicating that there is not an association between genotype and exposure length, visualized in Appendix A Figure A9. One benefit of GWECB is that both women and racial minorities were oversampled, allowing some examination of racial differences despite our modest sample size. The rs662 risk genotype is more common in those who identify as Hispanic or Black, non-Hispanic than in those who identify as White, non-Hispanic. Among White veterans, risk was higher with increased exposure in the GG and AG groups, but not in the AA group (Appendix A, Figure A10). The same is true among the Black and Hispanic veterans in the dataset (Appendix A, Figure A11), despite the smaller sample size. Additionally, we were unable to replicate a statistically significant interaction involving pesticides in our analysis, although prior work indicates that pesticide exposure may also interact with *PON1* rs662 genotypes [24,25,58,59,60,61]. Qualitatively the pesticide results show similar relationships with the SNPs in *PON1*. This is potentially attributable to the heterogeneity in the composition of pesticides experienced. The metabolism of carbamates (some pesticides and PB pills) is notably different than the metabolism of organophosphates (other pesticides and sarin gas) contributing to additional heterogeneity in the effect of the exposures. We used a variable that combined exposures from different pesticide sources into a variable indicating total exposure. Different pesticides may affect different biological pathways, and this further biologic heterogeneity may be reducing power to detect gene–environment interactions with pesticides. In addition, the GWECB measure of PB pills is based on self-report of exposure and did not query reaction to pesticide or PB exposure [62]. Finally, due to data and sample size limitations, we did not consider all potentially important exposures, such as chemical weapon exposure or oil-well fire smoke, or alternative timing of exposure effects, such as acute or chronic effects of exposures. A larger dataset or meta-analysis across similar analyses is required to address these limitations.

In summary, two significant signals were identified, associating GWI with the interaction between *PON1* and PB pill exposure. The primary result of the association between CDC Severe GWI and the rs662/PB pill exposure interaction corresponds with prior findings, as the rs662 locus is a functional variant that has been previously identified in smaller studies [14,30,38,40]. The secondary result of the association between respiratory symptoms and the interaction of PB with a trio of SNPs in high linkage disequilibrium (LD) with one another is a new finding and requires replication in additional, larger studies. The SNPs in these two signals do not appear to be in high LD with one another, although both are in *PON1* and both interactions are with PB pill exposure. We did not observe statistically significant interactions or marginal effects for any SNPs in *ACHE, BCHE* or *SOD1*.

## 5. Conclusions

Aligning with prior work in the area, this study identified an interaction between PB pill exposure and the G allele of rs662. Because this G allele codes for a functional variant, this interaction replicated prior work implicating genetic susceptibility to PB and pesticide exposure in GW veterans and identifies a potential biomarker for GWI risk.

## Figures and Tables

**Figure 1 brainsci-11-01558-f001:**
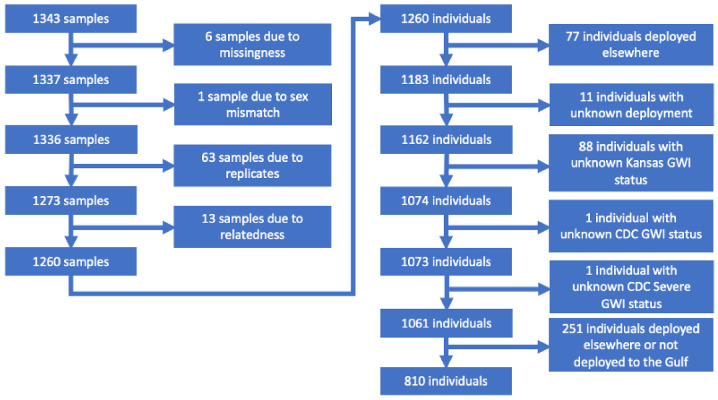
Block diagram: individuals in analytic dataset. Note: “Deployed to Gulf” throughout this manuscript indicates deployment to the Persian Gulf during the 1990–1991 Gulf War.

**Figure 2 brainsci-11-01558-f002:**
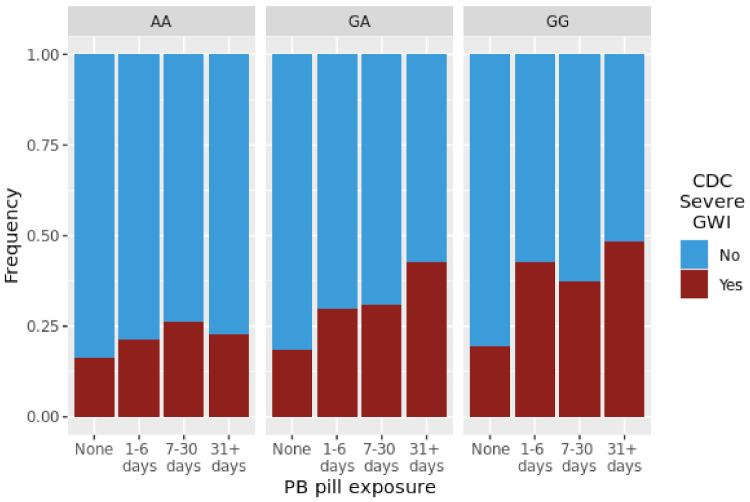
Proportion of individuals fulfilling the CDC Severe GWI case definition within each genotype-exposure group. Each block is labeled by genotype (AA, GA, GG), with each bar representing an exposure length.

**Figure 3 brainsci-11-01558-f003:**
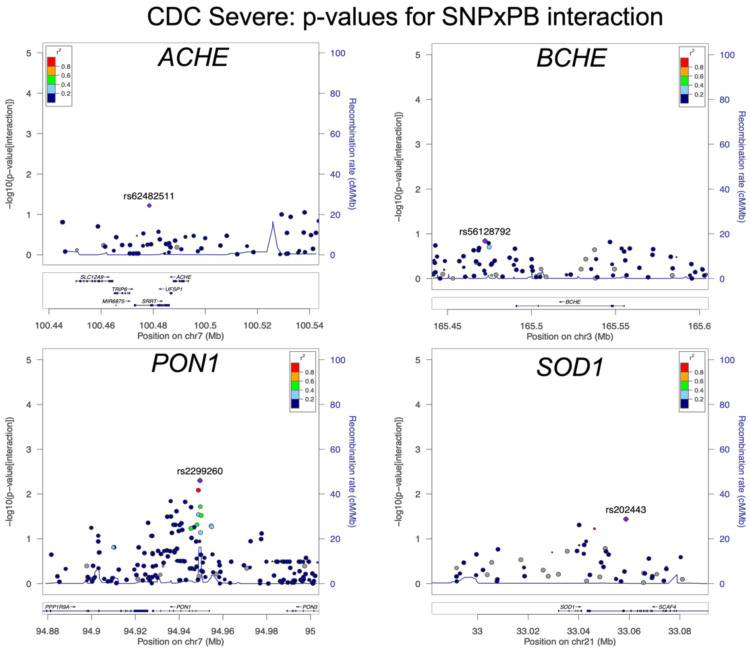
LocusZoom plots for CDC Severe GWI. Top SNP in each region is marked as a purple diamond, with linkage disequilibrium highlighted according to the *r*^2^ legend in upper right.

**Figure 4 brainsci-11-01558-f004:**
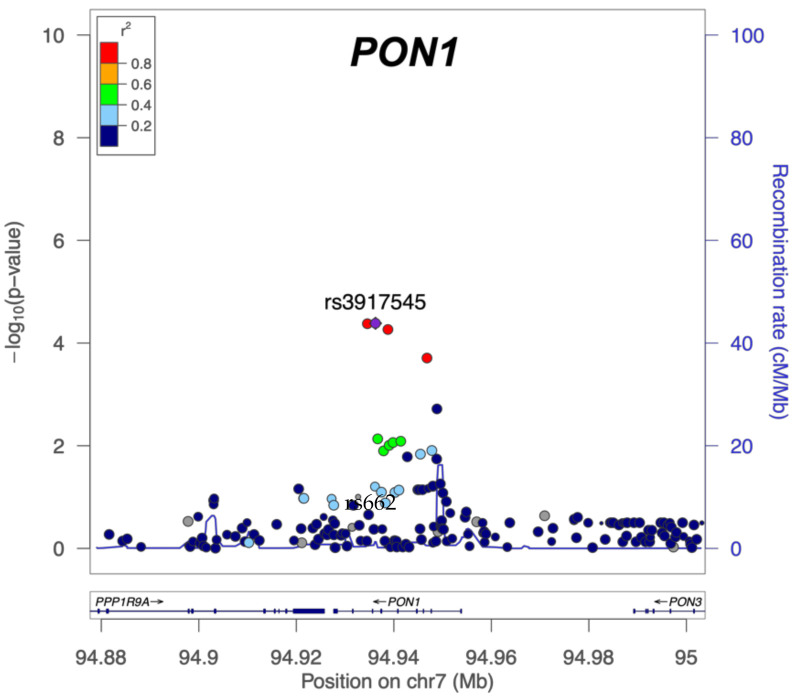
Kansas Respiratory Domain: *PON1*. Bonferroni corrected significance level of *p* = 0.00013 is 3.89 on the *y*-axis (−log10 scale). Three SNPs (purple and two red) are above that line. Rs662 is labeled for convenience (light blue dot).

**Table 1 brainsci-11-01558-t001:** Tier 1 candidate SNPs and Tier 2 SNPs in candidate genes.

Tier 1 SNPs: *BCHE*, *PON1*, and *ACHE* from Prior Studies				
**SNP**	**Gene**	**Variant**	**MAF**	**Citations**
rs1799807	*BCHE*	Atypical (A); lower catalytic rate; succinylcholine susceptibility	0.020	Zhu et al., 2020 [36], Goodall 2004 [49], Steele et al., 2015 [13]
rs662	*PON1*	192Q/R; functional variant of Pon1 Modifier for risk of sporadic ALS	0.396	Dardiotis et al., 2018 [50], Davies et al., 1996 [51], Haley et al., 1999 [14]; Verde et al., 2019 [39]
rs1799805	*ACHE*	H322N; Yt blood group;	0.037	Shapira et al., 2000 [43]
**Tier 2 SNPs: Remaining SNPs in *BCHE*, *PON1*, *ACHE*, and *SOD1***				
All SNPs in GWECB that are within 50 kb of the gene	Gene	Location (hg19)		Number of SNPs
	*BCHE*	Chr 3. 165490692–165555211 (+/− 50 kb)		84
	*PON1*	Chr 7. 94926988–94953844 (+/− 50 kb)		178
	*ACHE*	Chr 7. 100487615–100494614 (+/− 50 kb)		59
	*SOD1*	Chr 21. 33032006–33041244 (+/− 50 kb)		53

**Table 2 brainsci-11-01558-t002:** Demographics of deployed veterans by CDC Severe GWI case status, with column percentages.

Demographics		All (*n* = 810)	CDC Severe GWI	
			Yes (*n* = 212)	No (*n* = 598)
Sex	Female	180 (22%)	55 (26%)	125 (21%)
	Male	630 (78%)	157 (74%)	473 (79%)
Age Group	40–49	321 (40%)	98 (46%)	223 (37%)
	50–59	296 (37%)	74 (35%)	222 (37%)
	60+	193 (24%)	40 (19%)	153 (26%)
Race/Ethnicity	White, non-Hispanic	529 (65%)	101 (48%)	428 (72%)
	Black, non-Hispanic	137 (17%)	43 (20%)	94 (16%)
	Hispanic	76 (9%)	37 (17%)	39 (7%)
	Other or missing	68 (8%)	31 (15%)	37 (6%)
Income	Under $30,000	84 (10%)	33 (16%)	51 (9%)
	$30,000–59,999	179 (22%)	60 (28%)	119 (20%)
	$60,000–99,999	239 (30%)	65 (31%)	174 (29%)
	$100,000+	246 (30%)	35 (17%)	211 (35%)
Education ^^^	High school, GED, or less	75 (9%)	17 (8%)	58 (10%)
	Some college, or associate’s or bachelor’s degree	544 (67%)	161 (76%)	383 (64%)
	Advanced degree	166 (20%)	24 (11%)	142 (24%)
Service Branch^^^	Army only	366 (45%)	114 (54%)	252 (42%)
	Navy only	139 (17%)	28 (13%)	111 (19%)
	Air Force only	80 (10%)	12 (6%)	68 (11%)
	Marine Corps only	105 (13%)	26 (12%)	79 (13%)
	National Guard, any	81 (10%)	21 (10%)	60 (10%)
Military Component	Active Duty	470 (58%)	136 (64%)	334 (56%)
	Both Active Duty and Reserves	208 (26%)	59 (28%)	149 (25%)
	Reserves Only	127 (16%)	17 (8%)	110 (18%)
OEF/OIF Deployment ^a^	Yes	187 (23%)	38 (18%)	149 (25%)
	No	614 (76%)	168 (79%)	446 (75%)
Kansas Exclusionary Criterion	Yes	327 (40%)	112 (53%)	215 (36%)
	No	483 (60%)	100 (47%)	384 (64%)
rs1799807 (*BCHE*)	Homozygous reference (AA)	773 (95%)	206 (97%)	567 (95%)
	Heterozygous (AG)	36 (4%)	6 (3%)	30 (5%)
	Homozygous Atypical Variant (GG)	0 (0%)	0 (0%)	0 (0%)
rs1799805 (*ACHE*)	Homozygous reference (CC)	746 (92%)	199 (94%)	547 (91%)
	Heterozygous (CA)	63 (8%)	13 (6%)	50 (8%)
	Homozygous YT variant (AA)	0 (0%)	0 (0%)	0 (0%)
rs662 (*PON1*)	Homozygous 192Q variant (AA)	329 (41%)	67 (32%)	262 (44%)
	Heterozygous (AG)	340 (42%)	100 (47%)	240 (40%)
	Homozygous 192R variant (GG)	141 (17%)	45 (21%)	96 (16%)

^^^ Service Branch and Education do not add to 100%, as ‘missing’ or ‘unknown’ were not presented. ^a^ OEF/OIF deployment refers to Operation Enduring Freedom and Operation Iraqi Freedom; veterans who served in these conflicts served after 9/11.

**Table 3 brainsci-11-01558-t003:** Exposure time by CDC Severe GWI case status N (%).

Exposure	Length of Time	All	CDC Severe GWI	
			Yes	No
All		810	212 (26%)	598 (74%)
PB pill exposure	None	311	55 (18%)	256 (82%)
	Unclear or missing	138	41 (30%)	97 (70%)
	1–6 days	98	27 (28%)	71 (72%)
	7–30 days	114	35 (31%)	79 (69%)
	31+ days	149	54 (36%)	95 (64%)
Pesticide exposure	None	346	67 (19%)	279 (81%)
	Unclear or missing	106	37 (35%)	69 (65%)
	1–30 days	78	19 (24%)	59 (76%)
	31–62 days	124	37 (30%)	87 (70%)
	63+ days	156	52 (33%)	104 (67%)

**Table 4 brainsci-11-01558-t004:** Tier 1 candidate SNP results for CDC Severe GWI, three CDC Severe subdomains, Kansas GWI case definition and six Kansas subdomains. OR and *p*-values are from the interaction term of the logistic regression, adjusted for age, sex, Kansas exclusionary criterion, education, and income.

Outcome	Exposure	*ACHE*		*BCHE*		*PON1*	
		rs1799805		rs1799807		rs662	
		OR	*p*	OR	*p*	OR	*p*
CDC Severe GWI	PB	1.11	0.725	0.93	0.848	1.22	0.049 *
CDC Severe: Fatigue	PB	1.50	0.268	1.63	0.254	1.21	0.105
CDC Severe: Mood/Cognitive	PB	1.05	0.856	1.13	0.721	1.13	0.201
CDC Severe: Musculoskeletal	PB	0.99	0.953	0.86	0.639	1.13	0.202
Kansas GWI symptom criteria	PB	0.93	0.821	1.28	0.599	0.99	0.923
Kansas: Fatigue	PB	0.93	0.838	1.01	0.979	1.03	0.814
Kansas: Pain	PB	1.26	0.443	1.61	0.356	1.16	0.225
Kansas: Gastrointestinal	PB	0.96	0.883	1.18	0.614	0.96	0.663
Kansas: Respiratory	PB	1.15	0.581	0.95	0.868	1.18	0.080
Kansas: Mood/Neurological/ Cognitive	PB	0.58	0.251	1.09	0.855	1.21	0.278
Kansas: Skin	PB	0.59	0.042 *	1.3	0.408	1.01	0.897
CDC Severe GWI	Pesticides	0.67	0.179	0.78	0.548	1.07	0.485
CDC Severe: Fatigue	Pesticides	0.89	0.739	1.27	0.587	1.14	0.235
CDC Severe: Mood/Cognitive	Pesticides	0.65	0.123	1.17	0.676	1.00	0.958
CDC Severe: Musculoskeletal	Pesticides	0.61	0.063	0.95	0.887	1.04	0.684
Kansas GWI symptom criteria	Pesticides	0.60	0.128	2.11	0.189	1.05	0.660
Kansas: Fatigue	Pesticides	0.76	0.484	0.63	0.281	0.97	0.815
Kansas: Pain	Pesticides	0.67	0.151	1.47	0.362	1.17	0.185
Kansas: Gastrointestinal	Pesticides	0.84	0.490	1.11	0.747	0.82	0.029 *
Kansas: Respiratory	Pesticides	0.66	0.103	1.39	0.320	1.00	0.970
Kansas: Mood/Neurological/ Cognitive	Pesticides	1.09	0.893	1.72	0.353	1.11	0.562
Kansas: Skin	Pesticides	0.65	0.098	1.28	0.451	1.00	0.96

* Statistically significant with *p* < 0.05.

**Table 5 brainsci-11-01558-t005:** Top results for the Tier 2 SNP/PB interactions among the 384 tested SNPs across all GWI outcomes (*p* = 0.00013). From left to right, the top SNPs associated with: respiratory symptoms in interaction with PB pills (rs3917545), CDC Severe GWI in interactions with PB pill exposure (rs2299260), and CDC Severe GWI in interactions with pesticide exposure (rs62567349). Significant interactions are starred. Top interactions in each candidate gene are bolded.

Outcome	Exposure	*PON1*		*PON1*		*PON1*	
		rs3917545		rs2299260		rs62467349	
		OR	*p*	OR	*p*	OR	*p*
CDC Severe GWI	PB	1.45	3.20 × 10^−2^	0.68	5.00 × 10^−3^	1.39	5.50 × 10^−2^
CDC Severe: Fatigue	PB	1.54	2.50 × 10^−2^	0.64	7.60 × 10^−3^	1.52	3.10 × 10^−2^
CDC Severe: Mood/Cognitive	PB	1.25	1.70 × 10^−1^	0.82	1.20 × 10^−1^	1.23	2.10 × 10^−1^
CDC Severe: Musculoskeletal	PB	1.44	2.80 × 10^−2^	0.76	3.00 × 10^−2^	1.38	5.00 × 10^−2^
Kansas GWI symptom criteria	PB	1.23	3.20 × 10^−1^	0.99	9.70 × 10^−1^	1.23	3.20 × 10^−1^
Kansas: Pain	PB	1.28	2.30 × 10^−1^	0.73	3.50 × 10^−2^	1.31	1.90 × 10^−1^
Kansas: Gastrointestinal	PB	1.09	5.90 × 10^−1^	0.79	6.10 × 10^−2^	1.03	8.60 × 10^−1^
Kansas: Mood/Neurological/Cognitive	PB	2.30	3.30 × 10^−2^	0.89	5.60 × 10^−1^	2.18	4.40 × 10^−2^
Kansas: Fatigue	PB	1.14	5.70 × 10^−1^	0.95	7.90 × 10^−1^	1.13	5.80 × 10^−1^
Kansas: Respiratory	PB	2.00	**4.1 × 10^−5^ ***	0.78	5.50 × 10^−2^	1.86	2.00 × 10^−4^
Kansas: Skin	PB	1.11	5.20 × 10^−1^	0.92	4.80 × 10^−1^	1.11	5.10 × 10^−1^
CDC Severe GWI	Pesticides	1.45	2.20 × 10^−2^	0.89	3.70 × 10^−1^	1.54	8.60 × 10^−3^
CDC Severe: Fatigue	Pesticides	1.41	5.70 × 10^−2^	0.97	8.60 × 10^−1^	1.38	7.50 × 10^−2^
CDC Severe: Mood/Cognitive	Pesticides	1.05	7.70 × 10^−1^	0.94	5.90 × 10^−1^	1.07	6.50 × 10^−1^
CDC Severe: Musculoskeletal	Pesticides	1.19	2.40 × 10^−1^	0.86	2.20 × 10^−1^	1.26	1.40 × 10^−1^
Kansas GWI symptom criteria	Pesticides	1.30	1.80 × 10^−1^	0.85	2.70 × 10^−1^	1.37	1.20 × 10^−1^
Kansas: Pain	Pesticides	0.99	9.50 × 10^−1^	1.05	7.30 × 10^−1^	1.05	7.80 × 10^−1^
Kansas: Gastrointestinal	Pesticides	0.84	2.20 × 10^−1^	1.07	5.50 × 10^−1^	0.83	2.00 × 10^−1^
Kansas: Mood/Neurological/Cognitive	Pesticides	1.41	1.90 × 10^−1^	0.86	4.30 × 10^−1^	1.38	2.20 × 10^−1^
Kansas: Fatigue	Pesticides	0.99	9.60 × 10^−1^	0.79	1.40 × 10^−1^	1.03	8.80 × 10^−1^
Kansas: Respiratory	Pesticides	1.28	1.00 × 10^−1^	0.98	8.80 × 10^−1^	1.27	1.10 × 10^−1^
Kansas: Skin	Pesticides	1.04	7.90 × 10^−1^	1.09	4.90 × 10^−1^	0.99	9.20 × 10^−1^

## Data Availability

GWECB data may be requested by following instructions on the GWECB repository website: https://www.research.va.gov/programs/csp/585/repository.cfm (accessed on 4 May 2021).

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
