# Peer review of "Gene–Toxicant Interactions in Gulf War Illness: Differential Effects of the PON1 Genotype"

_brainsci, 2021, doi:10.3390/brainsci11121558_

Round 1

Reviewer 1 Report

This study by Vahey et al. sought to investigate candidate gene-toxicant interactions that may be associated with Gulf War Illness (GWI) utilizing data from the Gulf War Era Cohort and Biorepository (GWECB). Focusing on single nucleotide polymorphisms (SNPs) in or near ACHE, BCHE, and PON1 genes, which affect organophosphate (OP) hydrolysis, the authors report an interaction between the candidate SNP rs662 (PON1) and pyridostigmine bromide (PB) pills with CDC Severe GWI case status. They also reported an interaction between PB pill exposure, respiratory symptoms, and other SNPs in the PON1 gene (i.e. rs3917545, rs3917550, and rs2299255).  I have a number of concerns with the study’s methodology and conclusions:

First, the study’s main finding is an association between CDC Severe GWI and an interaction between rs662 (PON1)/PB pill use.  However, PON1 levels can vary at least 15-fold in the adult population (Davies et al., 2006; Roest et al., 2007; Furlong et al., 2007) and the catalytic efficiencies can vary significantly between the Q192R alloforms, depending on the substrate (Davies et al., 1996; Li et al., 2000). For this reason, studies that examined only PON1 SNPs tend not to be very informative (e.g., Jarvik et al., 2003). Haplotype analysis is also not particularly informative due to the lack of significant linkage disequilibrium across the PON1 locus (Jarvik et al., 2003).  Furthermore, several experienced PON1 investigators have pointed out the inadequacy of examining PON1 genotype alone as a risk factor for disease or exposure (e.g., Mackness et al., 2001; Deakin & James, 2004; La Du, 2003).  This is because many things that can happen between DNA and the final protein product. For example, studies have shown that are mutations in the PON1 gene that can result in the inactivation of one or the other PON1192 allele, based on discrepancies between PCR analysis and the functional PON1 status analysis (e.g., Jarvik et al., 2003).  Thus, PON1 status determinations tend to provide more information on both PON1 levels and the functional PON1192 alloform(s) present in plasma (Richter & Furlong, 1999; Jarvik et al., 2003).  The fact that this study only examined SNPs and not enzyme activity should be discussed as one of the limitations of the paper.

The authors reported an association between CDC Severe GWI and rs662 (PON1)/PB pill interaction.  However, PB is a carbamate compound, not an OP compound. The general consensus is that PON1 is not involved in the hydrolyzation of carbamates (e.g., Costa et al., 2013).  This should be addressed in the discussion.  On the other hand, BCHE is involved in the hydrolysis of both carbamates and OPs. As the authors noted in the introduction, Steele et al. (2015) previously reported an association between BCHE genotype and enzyme activity, PB exposure and GWI.  The authors should discuss potential reasons for their inability to replicate Steele et al.’s findings and the implications of their finding that genetic variants in an enzyme not known for hydrolyzing carbamate compounds had significant interactions with CDC severe GWI case status.

In this study, pesticide exposure during GW deployment was quantified by combining use of cream or liquid pesticides on the skin, wearing pesticide-treated uniforms, and using insect baits/no-pest strips in living area.  However, these deployment-related experiences likely reflect exposures to different types of pesticides.  For example, use of cream/spray pesticides likely reflects exposure to DEET, which is not an OP.  Wearing pesticide-treated uniform likely reflects exposure to permethrin, while use of insect baits/no-pest strips likely reflect exposure to methomyl, azamethiphos, and dichlorvos.  Studies have shown that different types of pesticides have different associations with GWI outcomes. When they are lumped together, there might appear to be associations with the “lumped” variable. But the actual associations would be obscured, and results would be misleading at best. Moreover, several of the pesticide variables tend to be highly correlated with one another, and also with PB use. Therefore, when evaluating pesticide associations with GWI or symptom outcomes, it is important to control for confounding that can result from these correlations.

This paper does not address another source of OP exposure that resulted during GW deployment: exposure to chemical warfare agents.

The 2014 IOM panel on GWI case definitions recommended the use of Kansas GWI definition for research purposes. Given that this study had the necessary data to ascertain Kansas GWI case status, it would be worthwhile to investigate/report the relationships between Kansas GWI case status and gene/GW-exposures.

In summarizing the demographics of the deployed veterans, Table 2 lists OEF/OIF deployment.  Does this refer to the veterans’ deployment in 2001 or later to the conflicts in Afghanistan and Iraq (Operations Enduring and Iraqi Freedom)?  Or did the authors mean to refer to Operations Desert Shield/Desert Storm in 1990-1991.  Similarly, in Figure 1 does “deployed to the Gulf” refer to Gulf War deployment in 1990-91 or subsequent deployments to Afghanistan and Iraq in 2001 or later?

Author Response

We thank the reviewer for their thorough and thoughtful comments.  We have uploaded our response as a file to best address their comments; please see attachment.

Reviewer 2 Report

  1. Page 2, Line 49- pyridostigmine is not an organophosphate. Please rewrite this section
  2. P2, L96-99 & P8, L242- It is not possible to make such a “blanket” statement due to the complexities of PON1 catalysis. The R version of the enzyme hydrolyses paraoxon, coumaphos-oxon more efficiently while the Q version is more efficient with diazoxon and sarin but both Q&R are equally efficient at hydrolysing chorpyriphos-oxon. Therefore labelling one as high activity and the other as low is both misleading and wrong as it is substrate dependent .
  3. P3, L136- In my copy appendix B does not contain a list of exposure questions and I can find no mention of appendix tables 1 & 2 or appendix C in the text. A corrected copy is required.
  4. P5, L204- what are OEF/OIF deployment?
  5. Discussion- The authors state in the abstract that the results ”indicate potential biological pathways”. However there is little or no detailed discussion of what these are/could be. It would be very useful to have the authors’ thoughts.

Author Response

We thank the reviewer for their thorough and helpful comments.  We have submitted a file to more completely respond to these comments; please see attachment.

Reviewer 3 Report

The authors investigate gene-toxicant interactions in Gulf war illness.  After a thorough review of the paper, I think it is well written and touches on an important subject. I have only minor edits

  1. Line 67-70: Cite more than one paper which speaks about the rodent models used in Gulf war illness where co-administration of PB and pesticides is used.
  2. Materials and Methods: Section 2.5; What specimen were used for genetic testing. Briefly describe how it was obtained, prepared and storage conditions etc
  3. Results: Section 3.2 (line 228): Your p value for is really boarder line, could a discussion of the odds ratio help in strengthening your argument?
  4. For section 3.2; Table 4. I suggest that the authors include a scatter plot(s) to show association between the GWI symptom and SNP in a particular gene where they have p values they have highlighted as significant. These are just 3 values and associated scatter plots can be included in the appendix of just under the table. I believe that these results are important, and must be made easy to read.

Author Response

We thank the reviewer for their thorough and thoughtful comments.  We have included a word document to more completely reply to each comment; please see the attachment.

Round 2

Reviewer 1 Report

The revised manuscript by Vahey and colleagues is improved; however, I still have some questions/comments for the authors:

1) The authors cite Haley et al. (1999) as previously showing that PON1 is associated with PB pills and Gulf War Illness.  However, this is not quite accurate.  First, Haley et al. classified Gulf War “Illness” by syndromes identified by factor analysis of symptoms. This is NOT the same as Severe CDC CMI or Kansas GWI. In fact, the general consensus in GWI research community is that the Haley syndromes are quite narrow and tend to underestimate the occurrence of GWI.  Moreover, Haley Syndrome 1 (Impaired cognition) is characterized by problems with attention, memory, and sleep along with depression. Syndrome 2 (Confusion/ataxia) is characterized problems with thinking and balance. Syndrome 3 (Neuropathic pain) is characterized by joint and muscle pain.  Given that respiratory symptom is NOT a characteristic of any of the three main Haley Syndromes, it would be worthwhile for the authors to discuss/explain why they think this particular symptom interacted significantly with PB and PON1 SNPs.  Might it be a spurious finding?

2) Haley et al. (1999) did not exactly show that PON1 is associated with PB pills and Gulf War Illness. Rather, Haley et al. reported that a history of advanced acute toxicity after taking pyridostigmine was correlated with low PON1 type Q arylesterase activity.  Acute toxicity after taking PB was identified by the GW veterans’ endorsements of 15 symptoms experienced after taking pyridostigmine tablets: Severe muscarinic side effects was assumed based on the veterans’ reports of excessive sweating, tearing, chest tightness, and nausea. Severe nicotinic side effects were assumed based on symptoms of muscle twitching and muscle cramps. My understanding is the current study did not look at/ask about the veterans’ reactions after taking PB pills. Instead, they only asked whether or not the veterans took PB pills during deployment and if so for how long.  If this is the case, then the distinction between this study's query of PB use and Haley et al.'s characterization of reaction after taking PB should be clarified in the manuscript.

3)  I still think it is important for the authors to discuss (or at least mention) in the paper that that PB is carbamate. While PON1 is capable of hydrolyzing the active metabolites of a number of OP, PB is NOT an organophosphate and NOT a known PON1 substrate.  For this reason, it would be informative for the the authors to discuss the reasons why they think CDC Severe CMI interacted significantly with PB and PON1 SNPs.

4) In justifying their reason for not examining other GW-related OP exposures (i.e., chemical warfare agents), the authors state that PB and pesticides were the only exposures that remained significantly associated with GWI after adjustments for other exposures.  However, in the paper the authors state that “Inhibition of AchE function is often cited as an explanation for why pesticides, PB, and nerve agents are suspected to be in the causal pathway of GWI…” and “Rodent models of GWI have been developed using combinations of pesticides, PB, sarin surrogate DFP, and chronic stress or corticosterone…”  Given that nerve agents have been proposed to be in the “causal pathway of GWI,” at the very least the authors should mention their rationale for excluding it from analyses in the manuscript.

5) The authors noted that the frequency of self-report of chemical warfare agent exposure was low in GWEBC. How was this exposure queried in GWEBC?  Often times veterans are not aware that they were exposed to chemical warfare agents. 

6) In describing PON1, the authors alternately refer to it as the Q/R version of the protein and as the A/G allele of the locus.  It is fine to explain which allele at the rs662 locus leads to which PON1 variant, but alternately referring to it as A/G and Q/R throughout the manuscript is confusing.

Author Response

The revised manuscript by Vahey and colleagues is improved; however, I still have some questions/comments for the authors:

1) The authors cite Haley et al. (1999) as previously showing that PON1 is associated with PB pills and Gulf War Illness.  However, this is not quite accurate.  First, Haley et al. classified Gulf War “Illness” by syndromes identified by factor analysis of symptoms. This is NOT the same as Severe CDC CMI or Kansas GWI. In fact, the general consensus in GWI research community is that the Haley syndromes are quite narrow and tend to underestimate the occurrence of GWI.  Moreover, Haley Syndrome 1 (Impaired cognition) is characterized by problems with attention, memory, and sleep along with depression. Syndrome 2 (Confusion/ataxia) is characterized problems with thinking and balance. Syndrome 3 (Neuropathic pain) is characterized by joint and muscle pain.  Given that respiratory symptom is NOT a characteristic of any of the three main Haley Syndromes, it would be worthwhile for the authors to discuss/explain why they think this particular symptom interacted significantly with PB and PON1 SNPs.  Might it be a spurious finding?

We thank the reviewer for their careful review of our work.  We have clarified the case definitions in lines 56-59.  We have added clarification of why the Kansas Respiratory Domain was tested (lines 117-120).  We have also clarified the need for replication and possibility of Type I error in the discussion section (lines 353-354, 393).

2) Haley et al. (1999) did not exactly show that PON1 is associated with PB pills and Gulf War Illness. Rather, Haley et al. reported that a history of advanced acute toxicity after taking pyridostigmine was correlated with low PON1 type Q arylesterase activity.  Acute toxicity after taking PB was identified by the GW veterans’ endorsements of 15 symptoms experienced after taking pyridostigmine tablets: Severe muscarinic side effects was assumed based on the veterans’ reports of excessive sweating, tearing, chest tightness, and nausea. Severe nicotinic side effects were assumed based on symptoms of muscle twitching and muscle cramps. My understanding is the current study did not look at/ask about the veterans’ reactions after taking PB pills. Instead, they only asked whether or not the veterans took PB pills during deployment and if so for how long.  If this is the case, then the distinction between this study's query of PB use and Haley et al.'s characterization of reaction after taking PB should be clarified in the manuscript.

Thank you for pointing out this important distinction.  We have added a note in the limitations that the exposure questions on the two questionnaires are different (lines 384-385).

3)  I still think it is important for the authors to discuss (or at least mention) in the paper that that PB is carbamate. While PON1 is capable of hydrolyzing the active metabolites of a number of OP, PB is NOT an organophosphate and NOT a known PON1 substrate.  For this reason, it would be informative for the the authors to discuss the reasons why they think CDC Severe CMI interacted significantly with PB and PON1 SNPs.

We thank the reviewer for their thoughtful comments both here and in the first round of reviews on the toxicology in this manuscript.  We discuss these results mainly focusing on the statistical genetics implications and have included a brief commentary on this in the discussion section (lines 379-380).

4) In justifying their reason for not examining other GW-related OP exposures (i.e., chemical warfare agents), the authors state that PB and pesticides were the only exposures that remained significantly associated with GWI after adjustments for other exposures.  However, in the paper the authors state that “Inhibition of AchE function is often cited as an explanation for why pesticides, PB, and nerve agents are suspected to be in the causal pathway of GWI…” and “Rodent models of GWI have been developed using combinations of pesticides, PB, sarin surrogate DFP, and chronic stress or corticosterone…”  Given that nerve agents have been proposed to be in the “causal pathway of GWI,” at the very least the authors should mention their rationale for excluding it from analyses in the manuscript.

We thank the reviewer for their careful reading of our work.  We have added lines 149-151 explaining this decision.

5) The authors noted that the frequency of self-report of chemical warfare agent exposure was low in GWEBC. How was this exposure queried in GWEBC?  Often times veterans are not aware that they were exposed to chemical warfare agents. 

We also find this an interesting problem with the exposure questionnaires.  The exposure was queried as Question 11g on the GWECB survey (Appendix Figure 1) reads as “Exposed to chemical or biological warfare agents.” We have not used this data, following RAC 2014 conclusions that pesticides and PB are causally associated with GWI, cited in lines 149-151.

6) In describing PON1, the authors alternately refer to it as the Q/R version of the protein and as the A/G allele of the locus.  It is fine to explain which allele at the rs662 locus leads to which PON1 variant, but alternately referring to it as A/G and Q/R throughout the manuscript is confusing.

We have streamlined this throughout the manuscript. 

Reviewer 2 Report

The manuscript has been extensively revised and restructured and is now much more coherant and "on point"

one very minor point L306 paraoxon misspelt

Author Response

We thank the reviewer for their thoughts and thorough read of our manuscript. We have fixed the spelling error.

Round 3

Reviewer 1 Report

The authors have adequately addressed my questions/concerns.